# The Colorectal Cancer Tumor Microenvironment and Its Impact on Liver and Lung Metastasis

**DOI:** 10.3390/cancers13246206

**Published:** 2021-12-09

**Authors:** Raghav Chandra, John D. Karalis, Charles Liu, Gilbert Z. Murimwa, Josiah Voth Park, Christopher A. Heid, Scott I. Reznik, Emina Huang, John D. Minna, Rolf A. Brekken

**Affiliations:** 1Division of Surgical Oncology, Department of Surgery, University of Texas Southwestern Medical Center, Dallas, TX 75390, USA; Raghav.Chandra@utsouthwestern.edu (R.C.); john.karalis@utsouthwestern.edu (J.D.K.); gilbert.murimwa@utsouthwestern.edu (G.Z.M.); josiah.vothpark@utsouthwestern.edu (J.V.P.); 2Hamon Center for Therapeutic Oncology Research, Simmons Comprehensive Cancer Center, University of Texas Southwestern Medical Center, Dallas, TX 75390, USA; john.minna@utsouthwestern.edu; 3School of Medicine, University of Texas Southwestern Medical Center, Dallas, TX 75390, USA; Charles.Liu@utsouthwestern.edu; 4Department of Cardiovascular and Thoracic Surgery, University of Texas Southwestern Medical Center, Dallas, TX 75390, USA; christopher.heid@utsouthwestern.edu (C.A.H.); scott.reznik@utsouthwestern.edu (S.I.R.); 5Division of Colon & Rectal Surgery, Department of Surgery, University of Texas Southwestern Medical Center, Dallas, TX 75390, USA; emina.huang@utsouthwestern.edu

**Keywords:** tumor microenvironment, colorectal cancer, colorectal liver metastasis, colorectal pulmonary metastasis, immuno-oncology, novel anticancer therapy

## Abstract

**Simple Summary:**

Colorectal cancer (CRC) is the third most common cancer worldwide. Metastasis to secondary organs, such as the liver and lungs, is a key driver of CRC-related mortality. The tumor microenvironment, which consists of the primary cancer cells, as well as associated support and immune cells, significantly affects the behavior of CRC cells at the primary tumor site, as well as in metastatic lesions. In this paper, we review the role of the individual components of the tumor microenvironment on tumor progression, immune evasion, and metastasis, and we discuss the implications of these components on antitumor therapies.

**Abstract:**

Colorectal cancer (CRC) is the third most common malignancy and the second most common cause of cancer-related mortality worldwide. A total of 20% of CRC patients present with distant metastases, most frequently to the liver and lung. In the primary tumor, as well as at each metastatic site, the cellular components of the tumor microenvironment (TME) contribute to tumor engraftment and metastasis. These include immune cells (macrophages, neutrophils, T lymphocytes, and dendritic cells) and stromal cells (cancer-associated fibroblasts and endothelial cells). In this review, we highlight how the TME influences tumor progression and invasion at the primary site and its function in fostering metastatic niches in the liver and lungs. We also discuss emerging clinical strategies to target the CRC TME.

## 1. Introduction

Colorectal cancer (CRC) is the third most common malignancy and the second leading cause of cancer-related mortality worldwide [1]. Metastasis is a primary driver of CRC-related mortality, with the liver and lungs representing the most frequently involved organs. While surgical resection of colorectal liver metastases (CRLMs) and colorectal pulmonary metastases (CRPMs) provides the only potentially curative treatment, select patients with unresectable metastatic CRC (mCRC) may benefit from other locoregional therapies, including radiofrequency ablation and stereotactic radiotherapy [2,3]. However, for most patients, systemic chemotherapy is the cornerstone of treatment. Recently, immune checkpoint inhibitors (ICIs) have been shown to have a significant clinical benefit in CRC that has high microsatellite instability or mismatch repair deficiencies [4]. This advance reflects the potential of tumor microenvironment (TME)-directed therapies for the treatment of mCRC. Here, we review the functions of the individual components of the TME in CRC, highlight the importance of the TME during the engraftment and growth of hepatic and pulmonary metastases, and discuss potential therapeutic implications of the TME in mCRC.

## 2. Limitations of Conventional Approaches for mCRC

Pulmonary metastasectomy and hepatic metastasectomy are the widely accepted first-line courses of management and the only cure for CRPM and CRLM, respectively [5,6,7,8]. The 2021 National Comprehensive Cancer Network guidelines recommend the resection of CRPMs in patients who have undergone curative (R0) resection of the primary tumor, have adequate cardiopulmonary reserve, have technically resectable disease, and have no extrathoracic metastases (with the exception of resectable CRLM) [5]. Similarly, for CRLM, those patients for whom an R0 margin is achievable with an adequate future functional liver remnant (20% in chemotherapy-naïve, noncirrhotic livers, and 40% for cirrhotic or post-treatment livers) are candidates for resection with curative intent [5,9]. A meta-analysis of 21 studies with 8361 patients demonstrated 5-year overall survival ranging between 24 and 82% after pulmonary metastasectomy [10]. Large-scale randomized controlled trials assessing the efficacy of pulmonary metastasectomy are lacking, and the only phase III randomized controlled trial (PulMiCC) comparing pulmonary metastasectomy versus active clinical monitoring for CRPM was stopped early due to poor recruitment [11]. A recent population-based Surveillance, Epidemiology and End Results Program study of 10,325 patients demonstrated no improvement in cancer-specific survival with pulmonary metastasectomy alone or in combination with hepatic metastasectomy on multivariable analysis [12].

Survival in CRLM is driven by the ability to completely extirpate hepatic disease, as positive margins incur an overall recurrence rate of 52% compared to 39% with R0 margins. Choti et al., Figueras et al., and Abdalla et al. [13,14,15] each demonstrated survival rates for CRLM patients approaching 60%, higher than earlier studies, despite more liberal indications for surgery. Unfortunately, up to 90% of mCRC patients present with unresectable disease, including up to 50% of patients undergoing metastasectomy [15,16,17,18]. Surgical trauma itself may confer a risk of recurrence. In lung resections, trauma to the nearby lung parenchyma has been associated with increased infiltration of myeloid cells and regulatory T cells into the TME, which may suppress CD8+ T-cell recruitment to the resection bed and increase the risk of recurrence [19]. Furthermore, the extent of resection and the magnitude of subsequent local tissue injury may be correlated to the immunosuppressive M2 polarization of local macrophages; a study of resection versus biopsy of oral squamous cell carcinomas in animal models demonstrated that resected lesions had higher levels of alternatively activated macrophages compared to biopsied lesions [20]. Thus, greater local trauma may be correlated with a favorable TME for tumor progression.

Chemotherapy alone has not been shown to be effective for CRLM or CRPM. A retrospective series of 2541 patients with mCRC demonstrated a 5-year overall survival of 10.8% for this approach. Of long-term survivors, only 2.2% (0.24% of the overall patient population) achieved a complete response to chemotherapy alone [21]. This underlies the importance of neoadjuvant and adjuvant systemic therapies, as well as continued research into novel therapeutics [17,22]. These mixed findings reflect the critical need to further investigate the role of the CRC TME on CRPM and CRLM pathogenesis and therapy.

## 3. Colorectal Liver Metastasis (CRLM)

The liver is the most common site of CRC metastases [23,24]. Up to 25% CRC patients may have synchronous CRLM, with an additional 50% developing CLRM during their disease course [25]. Oncogenotype may predict CRLM; Chu et al. [26] demonstrated that mutant *KRAS*^+^ CRC (found in 30–40% of cases) is associated with CRLM through the upregulation of YB-1 and insulin-like growth factor-1 receptor via the MEK-miR137 signaling pathway. Consistently, Brudvik et al. [27] identified that mutant *KRAS* status was inversely associated with overall and recurrence-free survival after CRLM resection in a meta-analysis of 1809 patients. The liver microenvironment represents a unique metastatic niche, and there may be differences in the immune landscape between liver metastases, the primary tumor, and other metastatic sites. Indeed, Tian et al. [28] demonstrated that CRC cells cultured on decellularized liver and lung scaffolds formed spheroid “metastases” in vitro and exhibited organ-specific tropism when injected into murine-models. Wei et al. showed in a 74-patient cohort that programmed death-ligand 1 (PD-L1) expression, a potential biomarker for ICI efficacy, and CD4 T-cell density were higher in CRLMs than the primary tumor [29]. An improved understanding of the CRLM TME, therefore, has implications on treatment strategy and may justify biopsy excisional biopsy of a CRLM to analyze both the metastasis and its microenvironment.

### 3.1. Pathophysiology of Colorectal Liver Metastasis

CRCs are thought to seed the liver primarily through the portal venous circulation. In contrast, cancer cells originating from outside of the gastrointestinal tract likely seed through the systemic circulation via the hepatic artery. These two sources of hepatic blood flow join at the point of entry into the liver sinusoids, where blood perfuses the liver parenchyma and returns to the systemic circulation via the centrilobular veins. Once circulating CRC tumor cells enter the liver sinusoids, the seeding of metastasis to the liver occurs in four dynamic, overlapping phases (Figure 1) [30,31,32].

#### 3.1.1. Microvascular Phase

Once in the portal vascular system, CRC circulating tumor cells pass through the portal venules into the sinusoidal capillaries. There, the cells encounter liver-specific defense mechanisms, and many are phagocytosed by stellate macrophages (Kupffer cells (KCs)), a process potentiated by natural killer cells (NKs) [33,34]. However, cells that escape innate immune surveillance must stop in the sinusoids to extravasate and form a metastatic lesion. While this mechanical arrest contributes to the adhesion of circulating tumor cells to the sinusoidal vasculature, intravital microscopy has shown that CRC circulating cells are usually smaller than the vessel diameter [35,36]. This suggests that specific cell-adhesion interactions can facilitate tumor cell arrest.

The liver sinusoids are lined by liver sinusoidal endothelial cells (LSECs), which express the critical cell-adhesion protein, E-selectin. E-selectin is a cytokine-inducible protein implicated in CRC cell adhesion to liver sinusoidal endothelial cells [37]. Notably, the inhibition of endothelial E-selectin has been shown to reduce the number of CRLMs up to 97% relative to the placebo in syngeneic murine models [37,38]. Furthermore, Khatib et al. [39,40] showed that CRC circulating cells trigger the release of proinflammatory TNF-α and IL-1β from KCs, thereby upregulating the expression of E-selectin and other adhesion molecules on liver sinusoidal endothelial cells and enhancing liver metastases [39,40].

#### 3.1.2. Extravascular/Pre-Angiogenic Phase

Once tumor cells have extravasated from the sinusoidal microvasculature into the space of Disse, hepatic stellate cells (HSCs) are activated by the pro-inflammatory cascade that was initiated in the microvascular phase. HSCs produce extracellular matrix (ECM) proteins, including collagen, laminin, fibrillin, and fibronectin, and provide a scaffold for tumor cells [41,42]. Additionally, KCs and neutrophils secrete matrix metalloproteinases (MMPs) and elastase, which degrade and remodel the ECM, facilitating tumor cell invasion [31,43]. Furthermore, cells in the TME have been implicated in the induction of an immunosuppressive niche, facilitating tumor cell growth. Huang et al. [44] demonstrated that HSCs co-cultured with T cells and CRC antigen-pulsed dendritic cells (DCs) significantly abrogated T-cell responsiveness and induced the expansion of immunosuppressive regulatory T cells (Tregs). In addition, portal injections of HSCs with tumor cells significantly increased CRLM compared to tumor cells alone [44].

#### 3.1.3. Angiogenic Phase

CRLMs are predominantly supplied by hepatic arteries [45]. The vascularization of liver metastases can occur through vascular endothelial growth factor (VEGF)-mediated angiogenesis or by the “co-opting” of existing vasculature. Importantly, a phase III trial by Hurwitz et al. [46] showed that the addition of bevacizumab, an anti-VEGF agent, to standard chemotherapy improved the median survival of patients with mCRC; however, the majority of patients did not exhibit a response to therapy [46]. This suggests the importance of other mechanisms of tumor vascularization independent of VEGF in some CRCs [42,47]. Notably, Frentzas et al. [48] described the co-opting of vessels in CRLM, a process by which cancer cells invade along the existing liver microvasculature, replacing hepatocytes and occupying the space adjacent to sinusoidal vessels. This commonly occurs in tumors that display a replacement growth pattern (see Section 3.1.4. below below) and is associated with resistance to anti-VEGF therapy. Importantly, the inhibition of actin-related protein 2/3 (Arp2/3), which is associated with the co-opting of vessels, was found to reduce vascular density in combination bevacizumab significantly more than bevacizumab by itself. Combination therapy also potently inhibited the growth of CRLMs with reduction in lesion area by approximately 75% compared to controls [48]. 

#### 3.1.4. Growth Phase

The growth of vascularized micrometastases leads to tumor cell proliferation and the establishment of clinically detectable metastases. Three histologic growth patterns have been described by Eefsen et al. [25] in a series of 24 patients with resected CRLMs: desmoplastic, pushing, and replacement. In the desmoplastic pattern, a band of fibrotic tissue separates tumor cells from the surrounding liver parenchyma. In the pushing pattern, liver cells are pushed aside by the growing metastatic lesion and assume a flattened morphology with no fibrotic band separating the invasive tumor front from the hepatic parenchyma. The replacement pattern describes tumor cell infiltration into the existing liver parenchyma, such that it supplants the parenchyma instead of simply pushing it away. In 83% of patients, identical growth patterns were identified in all metastases [25]. This classification is notable because the dominant replacement growth pattern is associated with a worse prognosis [49]. These differences indicate that different interactions of tumor cells, TME cells, and cells of the hepatic parenchyma lead to varying CRLM phenotypes.

## 4. Colorectal Lung Metastasis (CRPM)

The lungs are the second most common site of CRC metastasis. Studies suggest that 10–18% of rectal cancers and 5–6% of colon cancers metastasize to the lungs [50]. According to population-based studies, the incidence of synchronous CRPMs has been estimated to be up to 11%, and has increased over time [50,51,52]. Similar to CRLM, specific oncogenes are implicated in CRPM. Zhang et al. [53] demonstrated through extensive genomic profiling studies of clinical samples from five CRC patients that CRPM shared clonality with primary tumors. Furthermore, 14 of 27 mutated genes identified in CRPM (i.e., *KRAS*, *APC*, and *TP53*) were also found in the primary lesion, thus suggesting that these mutations likely existed in the primary lesion itself. As such, the identification of these mutations in the primary tumor during diagnostic workup or on post-resection pathologic analysis may be prognostically significant [53]. While surgical resection for isolated CRPMs is the only curative option for select patients, there are no high-quality clinical-trial data to guide practice. For most patients, systemic therapy is the cornerstone of treatment. Consistent with this, in their decellularized liver and lung scaffolding models, Tian et al. [28] found that CRPM-model cells were significantly more sensitive to chemotherapy compared to CRLM-model cells, suggesting that unique microenvironment features affect therapeutic response. As such, an improved understanding of the TME of CRPM may identify new therapeutic targets.

### Pathophysiology of Colorectal Metastasis to the Lungs

CRCs metastasize to the lungs through different mechanisms. Lymphatic vessels from the colon and proximal rectum drain into the inferior mesenteric nodes, which follows venous drainage into the portal circulation. Outflow from the distal rectum through the middle and inferior rectal veins drains into the internal iliac veins and directly into systemic circulation, thereby bypassing the portal system. Distal rectal lymphatics follow a similar course into the internal iliac nodes. As such, distal rectal tumors are more likely to present with early CRPM due to direct systemic drainage, as opposed to proximal tumors that must first filter through the liver [51]. Additionally, lymphatic vessels drain into the cisterna chyli and ultimately into the left subclavian vein through the thoracic duct, which may also lead to the development of CRPM [54].

The microenvironment of the lung parenchyma is enriched with a diverse array of immune cells that reside along the airway, including alveolar macrophages (dust cells), lymphocytes, and DCs, which are crucial for defense against airborne pathogens, toxins, and inflammatory substances [55,56]. Chronic inflammatory conditions, such as cigarette smoking and chronic obstructive pulmonary disease, alter the microenvironment in a way that lends itself to primary tumor development and the establishment of a pre-metastatic niche [57,58,59]. Indeed, current smoking status was found to be an independent risk factor for CRPM in a multivariable analysis of outcomes from 567 CRC patients [60]. 

Clearly, CRPMs (as well as other metastatic sites) represent an escape from immune surveillance. Thus, it is of great interest that a key feature of lung metastases is a tendency toward having a more immune responsive TME relative to metastases in other organs, such as the brain, liver, or bone [61]. García-Mulero et al. [61] analyzed resected tumor tissue and found that lung metastases from various primary tumors, including CRC, were enriched for genes associated with antigen presentation and immune effector cells, such as cytotoxic T lymphocytes (CTLs) and B-lineage cells. Moreover, they observed a low density of suppressor cells [61]. Lung metastases had a higher infiltration of CTLs, B-lineage cells, and DCs, but also expressed high levels of PD-L1 and cytotoxic T-lymphocyte–associated protein 4 (CTLA-4), which could partly explain their ability to escape immune surveillance [61]. However, infiltration by neutrophils, NK cells, and myeloid lineage cells was similar to other metastatic sites [61]. The overall high immunogenicity of the lung metastatic TME, as well as the elevated expression of PD-L1 and CTLA-4 in these lesions, may confer increased sensitivity to ICIs.

Altkori et al. [56] described a general mechanism for the formation of lung metastasis from extra-thoracic tumors. First, primary tumors secrete extracellular vesicles and pro-metastatic factors, including TGF-β, VEGF, and others that remodel the ECM, promote epithelial–mesenchymal transition (EMT), and facilitate invasion into the systemic circulation. These factors also promote recruitment of bone-marrow-derived cells into the microenvironment. Circulating tumor cells then extravasate into the pulmonary tissue. Type II alveolar cells recruit neutrophils, which suppress CTL activity and work with fibroblasts to facilitate seeding of tumor cells into the lung parenchyma. In addition, macrophages in the metastatic niche promote tumor cell survival and proliferation. Altogether, the established metastatic niche sustains tumor cell growth, promotes mesenchymal–epithelial transition, and stimulates angiogenesis [56].

## 5. The Tumor Microenvironment in CRC

To strengthen our understanding of the pathophysiology of CRLM and CRPM, it is crucial to first investigate in detail the function of individual components of the TME in CRC and their contribution to primary tumor invasiveness and subsequent metastasis to the liver and lung. The TME is the environment surrounding tumor cells and includes a complex array of immune cells, fibroblasts, endothelial cells, and stromal proteins. It is implicated in virtually every aspect of tumor progression and the metastatic cascade, including initial engraftment, growth, and immune evasion. Here, we review the function of selected cell types of the CRC TME (Figure 2).

### 5.1. Immune Cells

#### 5.1.1. Macrophages

Tumor-associated macrophages (TAMs) have historically been dichotomized as “M1-like” macrophages with pro-inflammatory, immunostimulatory, and anti-tumorigenic properties or “M2-like” macrophages with immunosuppressive and pro-tumorigenic properties [62]. In reality, TAMs exist on a spectrum between these two states; however, this traditional classification provides a useful framework for discussion. M1 macrophages produce pro-inflammatory immunostimulatory cytokines, including IL-1β, IL-6, and TNF-α [62,63]. Although the M1 phenotype is generally thought to be anti-tumorigenic, this notion is context-specific. For example, M1 TAMs can promote chronic inflammation in induced colitis models, and this is a risk factor for the development of CRC [64]. In a colitis-associated CRC model, Wang et al. [65] demonstrated that the densities of M1 and M2 macrophages vary throughout the inflammation–carcinoma sequence, but overall macrophage count increased as tumors became metastatic. Exposure of CRC cell lines to conditioned media from M1 TAMs decreased viability and promoted pro-apoptotic morphologic changes, while treatment with M2-derived conditioned media increased cell proliferation and expression of the anti-apoptotic markers survivin and BMI-1 compared to control conditioned media [66]. Activated M2 TAMs also contribute to tumor angiogenesis through secretion of VEGF along with other TME cells [67,68,69].

M2 TAMs promote invasion and metastasis [70,71,72,73,74]. TAM secrete of MMP-9, a key proteolytic enzyme that contributes to ECM remodeling in multiple cancers, including CRC [72]. Concordantly, Afik et al. [74] demonstrated that CRC TAMs were enriched for molecular signatures associated with ECM remodeling in transcriptomic and proteomic analyses [74]. M2 TAMs also contribute to the production of TGF-β, which potentiates ECM remodeling, EMT, and metastasis [71]. EMT pertains to tumor cell loss of epithelial features (i.e., the loss of apical–basal polarity, dissolution of tight junctions, etc.) and assumption of a mesenchymal-like phenotype [75]. The mesenchymal phenotype is associated with increased invasiveness and migratory potential [75]. The loss of E-cadherin expression, a hallmark of the epithelial phenotype, is associated with a poor prognosis in CRC [76]. TAMs also promote EMT through IL-6-mediated activation of the JAK2/STAT3 pathway and subsequent inhibition of the miRNA tumor suppressor miR-506-3p and its target, FoxQ1, as well as through secretion of CCL22 and subsequent activation of the PI3K/AKT pathway [73,77]. 

M2 TAMs also facilitate immune evasion through secretion of immunosuppressive cytokines IL-10 and TGF-β, which facilitate suppression of T lymphocytes [65,78]. Herbeuval et al. [79] demonstrated that TAMs induced an immunosuppressive environment through the secretion of IL-6, leading to STAT3-mediated production of IL-10 by CRC cells. Thus, M2 TAMs in the CRC TME drive tumor progression, invasiveness, and a more aggressive tumor phenotype. 

##### TAMs in CRLM and CRPM

TAMs are directly implicated in CRLM and may have context-specific pro-metastatic and anti-metastatic effects. In a microarray analysis of 159 resected CRLM specimens, the presence of CD68+ TAMs was associated with longer disease-free survival (DFS) on multivariable analysis [80]. Conversely, Wang et al. [81] demonstrated that CD206+ M2 TAMs were positively correlated with the presence of CRLM in patient samples; activation of the CXCL12/CXCR4 axis in CRC tumor cells promoted the exosome-mediated release of PTEN-suppressing miRNAs (miR-25-3p, miR-130b-3p, and miR-425-5p), which polarized TAMs to an M2 phenotype. Notably, murine tumors derived from HCT116 and MC38 CRC cells plus macrophages transfected with these miRNAs significantly increased tumor volume, and tail vein co-injection resulted in a significantly greater number and size of CRLM nodules compared to an injection of tumor cells alone [81]. KCs may also contribute to CRLM through the promotion of ECM remodeling and facilitation of tumor cell invasion for those CRC cells that escaped initial phagocytosis [43]. In this sense, they may serve a protective role against tumor cell infiltration earlier during metastasis. Indeed, Wen et al. demonstrated that the depletion of KCs at the earlier exponential growth phase increased CRLM burden in orthotopic murine models, while delayed KC depletion restricted tumor burden. This suggests that the pro- or anti-metastatic function of KCs may be time-dependent [82]. KCs and TAMs may also be impacted by CRLM resection. In a murine model of associated liver partition and portal vein ligation for staged hepatectomy for CRLM, mice who underwent this combined procedure had a significantly higher infiltration of KCs, which were polarized to the M1 phenotype, while TAMs were polarized to the M2 phenotype. This suggests that M1 KCs may be involved in a reactive inflammatory response, and that TAMs, perhaps M2-polarized as a consequence of induced hypoxia after the procedure, may contribute to persistent tumor progression and mixed therapeutic response [83].

How TAMs directly contribute to CRPM formation is not as well-elucidated. Cai et al. [71] demonstrated in CT26 CRC cell line–derived xenografts that the injection of TAMs into the tumor site significantly increased the number of CRPM nodules and total lung weight, a process dependent on TAM-derived TGF-β secretion and the induction of EMT through the modulation of the Smad2/3/4–Snail–E-cadherin pathway. Recently, *KRAS* mutational status was identified as a metastatic tropic factor for CRPM development, significantly more so than CRLM [84]. Patients with mutant *KRAS* tumors had a significantly shorter time to CRPM development and CRPM-free survival compared to those wild-type *KRAS* status [84]. Concordantly, Liu et al. [85] demonstrated that mutant *KRAS* reprograms macrophages to a M2 TAM-like phenotype through colony-stimulating factor 2 and lactate production, which promoted tumor progression and conferred EGFR inhibitor resistance. 

#### 5.1.2. Helper T Cells

CD4+ helper T cells are key mediators of the adaptive immune system, secreting cytokines to modulate the activity of other immune cells in response to infection or cancer. The presence of tumor-infiltrating lymphocytes (TILs) is associated with improved survival in CRC. In a study of 342 resected CRC specimens, high CD4+ T-cell density was associated with improved relapse-free survival and disease-specific survival [86]. However, CD4+ T-cell subtypes in the TME have varying impacts on tumor behavior. T-helper 1 cells (T_H_1) are traditionally thought to enhance CTL effector function and are instrumental for augmentation of the antitumor immune response [87]. Increased expression of T_H_1-cluster genes in resected CRC specimens is associated with improved disease-free survival, while higher numbers of T-helper 17 cells (T_H_17), which have an immunosuppressive function, are associated with poor survival [88]. This is consistent with prior preclinical studies, which suggest that T_H_17-derived cytokines, such as IL-17, IL-21, and IL-22, promote CRC tumor growth [89]. 

##### Helper T Cells in CRLM and CRPM

Kroemer et al. [90] demonstrated that the presence of expanded T_H_17 CD4+ T cells in resected CRLM was associated with poor prognosis. At the tumor site, while CD4+ T cells may be the most frequently encountered T-cell population, their ability to proliferate in response to tumor antigens after DC-mediated pulsing was significantly diminished [91]. These findings are consistent with those from Katz et al. [92], who demonstrated that high CD4+ T-cell density was inversely associated with survival after CRLM resection. In another study, Katz et al. [93] demonstrated through murine models that hepatic CD4+ T cells are generally polarized toward the T_H_2 phenotype and produced immunosuppressive cytokines. This may, in part, explain why the presence of CD4+ T cells negatively impacts prognosis. Little is known about how CD4+ T cells specifically impact CRPM. In other lung metastasis models including breast cancer, T_H_2-CD4+ T cells have been shown to augment metastasis through the potentiation of macrophage activity [94]. Further investigation into how CD4+ T cells specifically impact CRPM is necessary.

#### 5.1.3. Dendritic Cells (DCs)

DCs are professional antigen-presenting cells that are crucial for the initiation of the immune response to specific antigens through internalization of foreign antigens and subsequent presentation to T cells. To avoid immune surveillance, cancer cells may suppress DCs through multiple mechanisms, including the secretion of immunosuppressive TGF-β [95]. In CRC, myeloid DCs, the most common DC subtype associated with cell-mediated immunity and stimulation of naïve CD4+ T cells to the T_H_1 phenotype, are found in increased frequency at the tumor invasive front and are associated with lymph node invasion [96]. In contrast, Gulubova et al. [97] noted that advanced-stage CRC tumors had a lower density of CD83+ DCs in the stroma and invasive margin [97]. 

The mechanism through which these DCs enhance invasiveness has not been well-elucidated. Orsini et al. [98] demonstrated that DCs from CRC patients had impaired antigen-presenting capacity and reduced co-stimulatory molecule expression compared to DCs from healthy controls. Furthermore, CRC patient-derived DCs secreted increased levels of immunosuppressive IL-10 and decreased levels of immunostimulatory IL-12 and TNF-α [98]. Concordantly, Nagorsen et al. [99] demonstrated that while S100+ DCs were found more frequently in limited disease and were associated with improved survival, their presence in the CRC TME was also positively associated with Treg infiltration, further highlighting the paradoxical pro-tumorigenic function of CRC-hijacked DCs. Indeed, Hsu et al. [100] demonstrated that DCs harvested from CRC patients highly expressed CXCL1, which enhanced tumor cell migration, cancer cell stemness, and EMT. Thus, DC immunostimulatory capacity may be suppressed by CRC cells and modulated into an invasive, immunosuppressive phenotype, perhaps reflecting a reversion back to an immature phenotype [101]. 

##### DCs in CRLM and CRPM

Enhancing the DC presence in the CRLM TME may be a therapeutic target. Fused allogenic peripheral blood DCs with SW620 CRC cells promoted significantly higher secretion of IFN-γ by CD8+ T cells and, when injected into highly immunodeficient CRLM mouse models, markedly reduced tumor growth compared to sham-vaccinated CRC cells [102]. These findings are consistent with prognostic data suggesting that the absence of CD83+ DCs in resected CRLM specimens is associated with worse 5-year OS, partially due to diminished antigen presentation and immunostimulation [103]. This vaccination strategy was being investigated in a CRLM clinical trial which, despite being terminated early, demonstrated evidence of improved median DFS [104]. 

Hsu et al. [100] demonstrated that CXCL1 secretion by DCs augments CRPM. The sequencing of CXCL1-enriched SW620 CRC cells resulted in enrichment of prognostically significant genes including *PTHLH* (parathyroid hormone-like hormone). When these cells were implanted into murine livers, they were found to have a heightened capacity for CRPM formation compared to native SW620 cells, which was dependent on suppression of the p38/MAPK pathway and subsequent upregulation of *PTHLH* [105]. The inhibition of *PTHLH* selectively inhibited CRPM formation from CRLM, but not the growth of directly implanted cells or of liver metastases themselves [105]. 

#### 5.1.4. Regulatory T Cells (Tregs)

Tregs, characterized by CD25 and FoxP3 expression, are potent mediators of immunosuppression. Their presence in the TME is associated with increased metastasis and poor outcomes in numerous malignancies [106,107,108]. In their physiologic state, Tregs prevent autoimmunity and regulate the immune response by downregulating IL-2, releasing adenosine, and secreting immunosuppressive cytokines including TGF-β, IL-10, and IL-35 [109]. In CRC, the impact of Tregs in the TME is complex, and evidence suggests their function as pro- or anti-tumorigenic is context-specific. Ji et al. [110] demonstrated through mRNA profiling of 81 pretreatment biopsies of locally advanced rectal cancer patients that the Tregs were inversely correlated with prognosis. Conversely, other studies demonstrate that elevated FoxP3+ cell infiltration correlate with improved survival [86,111].

A potential explanation for the apparent contradictory effect of Tregs is likely due to their heterogeneity. Saito et al. [112] demonstrated that naïve Tregs (FoxP3^low^/CD45RA+) in CRC specimens could be stratified into terminally differentiated immunosuppressive FoxP3^high^/CD45RA-, and pro-inflammatory FoxP3^low^/CD45RA- subgroups. Compared to other tumors, such as melanoma, some CRC specimens had significantly higher numbers of FoxP3^low^/CD45RA- Tregs [112]. The authors therefore characterized CRC tumors as “type A” with a low expression of these inflammatory Tregs or “type B” with a high expression of inflammatory Tregs. Notably, type-B CRC demonstrated significant upregulation of genes involved with inflammation and immune response, including *IL-12*, *TNF-β*, and *TGF-β* [112]. Consistent with these findings, high *FOXP3* expression in type-A CRC tumors was associated with a worse prognosis [112]. 

##### Regulatory T Cells in CRLM and CRPM

While Treg heterogeneity has not been investigated in detail specifically for CRLM, their overall presence markedly impacts prognosis. Pedroza-Gonzalez et al. [91] demonstrated that CRLMs contain sequestered activated Tregs at the tumor site, which highly expressed glucocorticoid-induced TNF receptor (GITR) and were more potently immunosuppressive compared to Tregs from primary hepatocellular carcinoma. The ablation of GITR prevented Treg-mediated suppression of effector T-cell activity, suggesting a possible immunostimulatory therapeutic target [91]. Katz et al. demonstrated immunohistochemical analysis of 188 resected CRLM specimens that higher ratios of FoxP3+ Treg:CD4+ and FoxP3+ Treg:CD8+ ratios were associated with shorter 5-year overall survival compared to lower ratios (34% vs. 51%, and 35% vs. 46%, respectively) [113].

Tregs are prevalent in CRPM, but little is known about specific functions in this niche [114]. Resident Tregs may contribute to the pulmonary pre-metastatic niche. In physiologic states, pulmonary tissue is enriched with Tregs, which are crucial in regulating immune responses to inhaled environmental antigens. Through many of the previously discussed mechanisms, pulmonary Tregs may promote immunosuppression in the lung microenvironment and foster a niche for pulmonary metastasis [115]. 

#### 5.1.5. Neutrophils

Neutrophils are the first responders to acute infection or inflammation [116]. They secrete lytic enzymes, phagocytose certain microbes, generate neutrophil extracellular traps, and secrete numerous cytokines that sustain the inflammatory response and recruit other immune cells as part of the innate immune response. Tumor-associated neutrophils (TANs) are associated with prognosis; as such, understanding the function of TANs is an emerging area of investigation [117]. Fridlender et al. [118] demonstrated that TGF-β production in the CRC TME polarizes TANs from an immunostimulatory “N1” phenotype to a pro-tumorigenic “N2” phenotype. TANs promote invasiveness through modulation of angiogenesis and resistance to VEGF inhibition. In a colitis-associated CRC mouse model, Itatani et al. [119] demonstrated that the induction of colitis increased granulocyte colony stimulating factor (G-CSF) levels in anti-VEGF therapy-resistant tumors, which was associated with increased Bv8/PROK2+ neutrophil infiltration. The dual inhibition of G-CSF or its receptor along with VEGF inhibition resulted in significantly suppressed tumor formation and angiogenesis [119]. 

TANs also contribute to immune evasion. In an inducible CRC murine model, Germann et al. [120] demonstrated that presence of T cells in adenomas was diminished and inversely correlated with neutrophil infiltration. These TANs were enriched for immunosuppressive gene expression and secreted MMP-9, which suppressed T-cell proliferation through the activation of TGF-β [120]. Overall, TANs may also facilitate the dissemination of tumor cells by augmenting degradation of the basement membrane, which enhances tumor mobility and promotes tumor cell extravasation [121].

##### TANs in CRLM and CRPM

TANs contribute to the growth of CRLM through multiple mechanisms. In murine models, Gordon-Weeks et al. [122] demonstrated that CRLM-associated neutrophils were enriched in tumors, expressed high levels of fibroblast growth factor-2 (FGF2), and enhanced the growth of CRLM in mice. The depletion of neutrophils 7 days after an intrasplenic injection of CRC cells diminished CRLM development and angiogenesis. Monoclonal antibody (GAL-F2) blockade of FGF2 also restricted tumor growth and normalized vasculature, suggesting an FGF2-dependent mechanism through which TANs contribute to CRLM [122]. This is consistent with the previously discussed findings of Itatani et al., who also demonstrated that granulocyte-colony stimulating factor (G-CSF) secretion by CRC cells enhanced TAN recruitment in CRLM as well as in primary CRC tumors, and conferred similar resistance to anti-VEGF therapy [119]. Additionally, Yang et al. [123] demonstrated that neutrophils derived from CRC patient sera prominently secreted neutrophil extracellular traps (web-like consolidations of chromatin and proteases thought to be released by activated neutrophils and believed to enhance tumorigenesis perhaps through trapping cells at metastatic sites), which correlated with the risk of CRLM. Furthermore, CXCL8 secretion from CRC cells mediated the production of neutrophil extracellular traps, which enhanced CRLM formation in vivo [123].

A combination of SMAD4 deficiency in CRC cells plus CCR1+ TANs is implicated in the progression of CRPM [124]. SMAD4 is a tumor suppressor, and its inactivation is associated with multiple cancers, including CRC [125,126,127]. *SMAD4*-deficient CRC cells secrete higher levels of CCL15, which recruits CCR1+ TANs through the CCL15-CCR1 chemokine axis to promote CRPM [124]. TANs are also critical drivers for the development of the pre-metastatic niche at secondary sites like the lung [128,129,130]. In a model of melanoma pulmonary metastasis, El Rayes et al. [59] demonstrated that LPS-inflamed lungs were enriched for Ly6G+ neutrophils and developed a greater metastatic burden compared to controls. These TANs proteolytically degraded thrombospondin-1 (Tsp-1), an ECM protein that regulates inflammation, inhibits angiogenesis, and restricts tumor growth [59]. 

#### 5.1.6. Cytotoxic T Cells

CTLs are the primary effector cells of the adaptive immune response and are a crucial component of antitumor immunity. The presence of CTLs in patient-derived CRC tumor cores is associated with a reduced risk of recurrence, particularly in higher-stage tumors [131]. Concordantly, the CRC Immunoscore prognostic tool was developed to supplement the TNM staging paradigm and is based on the prevalence of infiltrating CD3+ cells and CTLs. Higher Immunoscore values positively correlate with improved survival and are corroborated by internal and external validation studies [132,133]. 

To promote immune evasion, CTL activity is suppressed through multiple mechanisms. O’Malley et al. [134] demonstrated that TNF-α–mediated inflammation in CT26 CRC cells induced PD-L1 expression in stromal cells, resulting in the inhibition of activated granzyme-secreting CD8+ T cells [134]. Tumor and stromal cells express PD-L1, which, upon binding to the PD-1 receptor on T cells, suppresses T-cell activation. In CRC, a higher expression of PD-L1 is associated with higher stage, higher grade, lymph node involvement, distant metastasis, and reduced overall survival [135,136,137]. The PD-1/PD-L1 axis is an attractive target to increase immune targeting. Indeed, PD-1 blockade with pembrolizumab was FDA-approved for MSI-high/mismatch repair-deficient subtype advanced tumors in the pivotal KEYNOTE 177 trial [4,138]. Other TME cells also curtail CTL function through TGF-β secretion, which inhibits CTL expression of the key cytolytic enzymes: perforin, granzyme, and Fas ligand [139]. Furthermore, Tregs recruited via the CCL5/CCR5 signaling pathway have a heightened ability to kill CTLs in CRC murine models [140].

##### CTLs in CRLM and CRPM

CRLMs are more highly enriched for PD-L1 expression compared to primary tumors, particularly rectal cancer liver metastases [29]. In a study of 74 patients with resected CRLM and primary tumors, Wei et al. [29] found that high PD-L1 expression was significantly associated with an enrichment of CD4+ T cells and CTLs. Concordantly, Katz et al. [92] observed that a high density of CTLs positively correlated with 10-year survival after CRLM resection, and on multivariable analysis, a high CD8+/low CD4+ phenotype was significantly associated with better long-term survival. 

Similarly, Remark et al. [141] found that an increased presence of LAMP+ DCs and CTLs was positively associated with improved overall survival in CRPM, but not in renal cell carcinoma lung metastases, suggesting a context-specific function. Furthermore, the density of CTLs, as well as DCs and NKs, in primary CRC tumors correlated with densities in CRPM, suggesting a reproducible immune pattern from the primary tumor possibly secondary to TME “imprinting” from the primary site or the development of educated immune cells [141]. The authors suggest that, since tertiary lymphoid structures are prevalent in CRPM, these T cells may be more educated compared to renal cell carcinoma metastases, which tend to have a scarce presence of tertiary lymphoid structures [141]. This may point toward a sensitivity of CRPM and CRLM to ICIs, particularly in patients with MSI-high/mismatch repair-deficient subtype advanced CRC [4,138]. 

### 5.2. Stromal Cells and the Extracellular Matrix

#### 5.2.1. Extracellular Matrix

The ECM comprises structural proteins, including collagen, proteoglycans, hyaluronic acid, and glycosaminoglycans, whose structure and function can be remodeled by cells in the TME [142]. In addition to being the physical scaffolding for tumor cells, the ECM dynamically contributes to cell–cell adhesion, paracrine signaling, tumor proliferation, immune evasion, and metastasis [143]. Remodeling of the ECM is a consistent feature of the TME in many indications including CRC. Yang et al. [144] demonstrated that density of the remodeling enzyme MMP-9 was markedly greater in CRC compared to normal mucosa and higher expression was associated with worse prognosis. MMP-2 has also been implicated in ECM degradation and correlates with lymphatic invasion and advanced stage [145,146]. MMP-2 and MMP-9 encode collagenases that target type IV collagen, which is found in the basement membrane; and the degradation of type IV collagen can enhance tumor cell motility and invasion [145,147,148]. The biomechanics of the ECM also contribute to CRC proliferation. Using atomic force microscopy, Brauchle et al. [149] demonstrated that collagen-rich regions of the CRC ECM were stiffer compared to normal tissue. Stiffness of CRC has been associated with metastasis and EMT [150]. Li et al. [143] observed that the density of type I collagen positively correlated with stage, while type IV collagen density was reduced in higher stage tumors. This latter finding is consistent with invasion through the basement membrane [151]. The inverse relationship between density of type IV collagen and CRC stage likely reflects increased ECM remodeling, partly through MMP-2 and -9 activity [147]. At distant sites, proteomic analyses of control vs. matched CRLM patient tissues suggest that, while the ECM of CRLM more closely resembles the primary tumor compared to liver parenchyma, it is enriched for several unique proteins (i.e., SPP1 and COMP) [152]. Yuzhalin et al. [153] found that the CRLM ECM had higher levels of citrullinating enzyme peptidylarginine deiminase 4 (PAD4) compared to benign liver tissue, primary CRC, or colonic mucosa. CRC cells grown in citrullinated type I collagen exhibited mesenchymal-to-epithelial transition (reversion to a proliferative epithelial phenotype of tumor cells) in vitro and enhancement of CRLM growth in vivo [153].

#### 5.2.2. Cancer-Associated Fibroblasts

Cancer-associated fibroblasts (CAFs) contribute significantly to ECM maintenance, desmoplasia, angiogenesis, immunosuppression, invasion, and chemoresistance [154,155]. While a majority of CAFs arise from resident stromal fibroblasts, mesenchymal stem cells (MSCs) and mesothelial cells also contribute a significant proportion of the CAF progenitor population [154,156]. Bone-marrow-derived MSCs, which are pluripotent stem cells involved in tissue remodeling, may be recruited to tumor sites to aid in tumor growth and progression [157]. Once localized at the tumor site, MSCs are induced by tumor-derived factors to assume a CAF-like phenotype [158]. Meanwhile, mesothelial cells contribute to the CAF population in highly invasive tumors via mesothelial-to-mesenchymal transition, which has been implicated in CRC peritoneal carcinomatosis [159]. Overall, in CRC, two major subpopulations of CAFs have been identified through single-cell RNA sequencing: CAF-A (which express ECM remodeling-associated genes, including *MMP2*, *DCN*, and *COL1A2*) and CAF-B (myofibroblast-like that express *ACTA2*, *TAGLN*, and *PDGFA*) [160].

CAFs promote immune evasion [120,154,161,162]. In microsatellite stable CRC samples, Tauriello et al. demonstrated an inverse correlation between the immunostimulatory T_H_1 to naïve T-cell ratio and the mean expression of CAF-specific genes [162]. Tumor invasive margins had high levels of stromal TGF-β, of which CAFs were the primary source. The elevated expression of TGF-β in the TME inhibited T_H_1 T-cell function. Additionally, CAFs secrete CXCL8, which attracts monocytes to the CRC TME and promotes M2 polarization, further contributing to immunosuppression [163].

CAFs promote metastasis through ECM remodeling and EMT in multiple cancers, including CRC [164,165,166,167]. Their presence in the CRC intratumoral stroma is associated with lymphatic invasion [168]. Through the secretion of ECM remodeling enzymes, collagen, and other cytoskeletal proteins, CAFs promote desmoplasia, which can be identified in up to 78% of CRC tumor specimens [168]. TGF-β/Smad2 signaling is prominently activated in CAFs, and CRC cells themselves can stimulate TGF-β production in CAFs with resulting expression of α-SMA and differentiation to a myofibroblastic phenotype [169]. These upregulate the expression of invasion-related proteinases, including multiple MMPs [169]. Notably, TAMs may regulate CAF activity in the CRC ECM with respect to the expression of collagen types I and XIV [74]. The expression of collagen crosslinking enzyme lysyl oxidase–like 2 (LOXL2) by CAFs in CRC tumors is associated with a higher rate of recurrence and worse overall survival and disease-free survival [170]. Through LOXL2, CAFs also stimulate EMT through activation of the FAK pathway, with a resultant reduction in E-cadherin protein expression. CRC cells co-cultured with these activated fibroblasts demonstrated increased migration rates [167]. CAFs also promote angiogenesis through secretion of IL-6; Nagasaki et al. [171] showed that CRC cells potentiated IL-6 secretion from CAFs with resulting upregulation of *VEGFA* expression. Thus, CAFs contribute to CRC progression through immunosuppression, ECM remodeling, and the promotion of EMT.

##### CAFs in CRLM and CRPM

In murine models, CRLM tumor cells recruit CAFs to the metastatic site, which contributes to tumor progression [172]. Circulating levels of TGF-β are a predictor of future development of CRLM after resection of the primary tumor [173]. Calon et al. [174] demonstrated that increased stromal TGF-β signaling significantly promoted CRPM and CRLM in murine models through the upregulation of GP130–p-STAT3 signaling in CRC cells. TGF-β–stimulated CAFs in the CRC TME secreted IL-11, a key ligand for GP130, resulting in the upregulation of anti-apoptotic factors MCL-1 and Bcl-2 [174]. Intrasplenic injection of HT29-M6 cells autonomously producing IL-11 more robustly colonized the liver and exhibited reduced apoptosis within the first hours of colonization compared to controls [174]. CAFs also potentiate metastasis through secretion of hepatocyte growth factor (HGF). Zhang et al. [175] demonstrated that secretion of HGF by CAFs upregulated CD44, and enhanced adhesive and migratory capacity of CRC cells in vitro and increased CRLM and CRPM formation in vivo. Resident fibroblasts also contribute to CRLM formation as well as the TME CAF population. As discussed, activated HSCs secrete ECM components and provide scaffolding for newly seeded CRC cells and contribute to immune evasion [44]. Furthermore, HSC-derived myofibroblasts secrete stromal-cell-derived factor 1 (a CXCR4 ligand), which can promote primary tumorigenesis in nude mice. Concordantly, CXCR4 may be a therapeutic target, as its inhibition with AMD3100 significantly diminished CRLM [176]. HSCs may not be the primary progenitors for CRLM CAFs, but rather resident portal fibroblasts. Mueller et al. [177] demonstrated that CRLM CAFs are similar to resident portal fibroblasts in their myofibroblastic phenotypes (α-SMA+, ICAM-1+, Thy-1+) and do not closely resemble HSCs (in contrast to the HSC phenotype, they are negative for glial fibrillary acidic protein, desmin, and neural cell adhesion molecules). 

CAFs impact the prognosis for CRPM. In a study of 181 CRC-patient primary-tumor and metastatic-site specimens, Kwak et al. [178] demonstrated that CRPM had lower density of CAFs compared to primary tumors, but similar to the density at other tumor sites. The presence of CAFs at these metastatic sites and the loss of their PTEN expression were significantly associated with a worse prognosis [178]. CAFs in the CRPM TME prominently secrete heat-shock protein-27 (Hsp27), a protein implicated in angiogenesis, EMT, and fibroblast motility [179]. Hsp27 expression was associated with reduced disease-free survival and overall survival after CRPM metastasectomy [179]. 

#### 5.2.3. Endothelial Cells

Tumor vasculature prominently impacts CRC invasion and is an established therapeutic target [180,181]. Endothelial cells (ECs), along with pericytes, smooth muscle cells, and progenitor cells, comprise the vasculature [182]. While EC generally remain quiescent in healthy tissue, malignancy can induce vascular turnover and cell proliferation [183]. Local injury, hypoxia, and rapid malignant proliferation initiate paracrine signaling networks where factors such as VEGF, PDGF, CXCL8, and TNF-α secreted by multiple TME cells drive neovascularization to perfuse proliferating tumor tissue [184]. ECs are implicated in invasion and metastasis. Zhang et al. [185] demonstrated that expression of vascular cell-adhesion molecular 1 (VCAM1) expression in CRC was associated with more invasive features and poorer prognosis. Forced expression of VCAM1 in CRC cells promoted pseudopodia formation and increased trans-endothelial migration in vitro. Furthermore, after CRC cells have entered the systemic vasculature, circulating tumor-derived ECs have been recently identified that may reflect a unique EC state that contributes directly to tumor progression and metastasis [186]. 

##### Endothelial Cells in CRLM and CRPM

As discussed, LSECs are central in early metastatic seeding, which is driven by E-selectin mediated adhesion to the endothelium and VEGF-mediated lymphangiogenesis [37,187]. LSEC lectin (LSECtin) promotes migration in malignant colorectal cell lines in vitro and in vivo and inhibition of LSECtin reduced CRLM formation [188]. Additionally, tumor-activated LSECs secrete ICAM-1, which drives IL-1 production and decreased antitumor immune activity through impaired IFN-γ production and mannose receptor-dependent endocytosis [189]. LSECs have also been shown to induce chemotaxis and outgrowth in CRC by secreting macrophage inhibitory factors, whose levels correlate with CRLM development [190].

In CRPM, CUB domain-containing protein 1 (CDCP1) is a cell-surface marker which identifies colorectal circulating stem cells that exhibit tropism to the lung [191]. CDCP1 expression is associated with increased risk of CRPM formation and inferior metastasis-free survival, and it may be implicated in the seeding and adhesion of CTCs to the pulmonary vasculature [191]. Elevated VEGF expression in CRPMs has been identified as a prognostic marker which predicts lack of benefit from CRPM resection [192]. 

**Figure 2 cancers-13-06206-f002:**
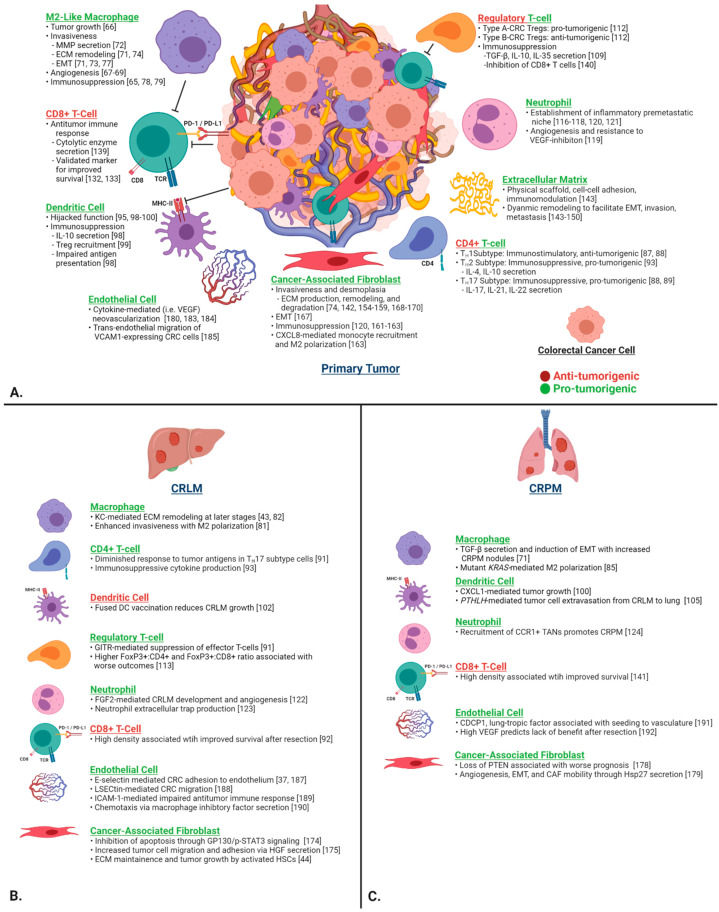
(**A**) Contributions of the key individual components of the CRC TME on tumor progression, invasion, ECM remodeling, immunosuppression, and metastasis in the primary tumor. (**B**) CRLM- and (**C**) CRPM-specific functions are depicted in the lower two panels. Green headings signify pro-tumorigenic functions, and red headings signify anti-tumorigenic functions. Abbreviations: CRC, colorectal cancer; ECM, extracellular matrix; EMT, epithelial–mesenchymal transition; MMP, matrix metalloproteinase; TME, tumor microenvironment.

## 6. Implications of TME Factors on the Treatment of mCRC

We have thus far reviewed the contribution individual cellular components of the CRC TME have on tumor progression, metastasis, and prognosis. Understanding these functions may be crucial for understanding how they impact the efficacy of established and emerging and novel therapies. Below, we highlight some of the key advancements made in the treatment of CRC and how they are influenced by the TME. Additionally, a summary of meta-analyses examining the efficacy of surgery and local ablative therapies for the treatment of CRPM and CRLM is provided in Table 1. 

### 6.1. Local Ablative Techniques

#### 6.1.1. Stereotactic Body Radiation Therapy

Patients who are not candidates for hepatic metastasectomy or pulmonary metastasectomy may be managed with stereotactic body radiation therapy (SBRT), through which focused radiation is used to ablate tumors while minimizing nearby tissue damage. A meta-analysis of 18 studies showed that SBRT for CRPM resulted in 3-year local control, overall survival, and progression-free survival rates of 60%, 52%, and 13%, respectively [193]. Rectal cancer pulmonary metastases may be more radioresistant, as primary rectal tumors are frequently treated with neoadjuvant chemoradiation; this may select for the survival of radioresistant cells or higher rates of *KRAS* mutations, which are associated with poorer survival after SBRT [194]. Genetic studies have also identified CRPMs as more intrinsically radioresistant [193]. Despite this, SBRT may enhance the immunogenicity of CRPMs. For example, in primary lung cancer, SBRT has been demonstrated to prime the immune TME through the upregulation of immunogenic cell surface markers (i.e., ICAM-1, MHC-1, or Fas), increase the release of tumor antigens, and enhance immunostimulatory cytokine production. Through the abscopal effect, even distant metastases not primarily targeted by radiation may exhibit enhanced antitumor immune response and cell death, perhaps in response to the “vaccine effect” of SBRT, with the release of CD8+ T cells stimulated by danger signals and cytokines. Conversely, SBRT increases the release of TGF-β and increases PD-L1 expression, which may simultaneously increase immunosuppression [195]. To this end, a phase Ib/II trial investigating fusion blockade of TGF-β and PD-L1 (M7284) and radiation therapy for MSI+ mCRC [196] and a phase Ib trial investigating SBRT + pembrolizumab for CRLM are ongoing [197]. 

#### 6.1.2. Radiofrequency Ablation

Radiofrequency ablation therapy (RFA) relies on the insertion of electrodes into a tumor and passing high-frequency electrical currents to heat and destroy cells [15,198]. It is an established approach for CRLM with 5-year survivals ranging from 20 to 48.5% [198]. It has also been shown to be safe and effective for inoperable CRPM, though studies are limited by study size and patient heterogeneity [199,200]. According to a large meta-analysis, the 1-year overall survival following RFA ranges between 83.9 and 95%, while the 3-year survival ranges between 46 and 59.6% [199]. Unfortunately, recurrence at the ablative site is a persistent concern. This may be due, in part, to the modulation of the immune microenvironment. Stimulation of the antitumor immune response by RFA may contribute to its effectiveness but may also trigger reactive immunosuppression. In a case-controlled study of 38 patients with synchronous CRLM who underwent neoadjuvant RFA prior to hepatic metastasectomy, Shi et al. [201] demonstrated that RFA significantly increased the number of infiltrating T cells with a higher CD8+/CD4+ ratio. Interestingly, PD-L1 expression on tumor cells and tumor-associated immune cells was also upregulated, suggesting a “self-limiting” phenomenon, which was further corroborated by murine studies that demonstrated a decrease in the initially high CD8+/Treg ratio after RFA as well as a failure to generate a durable suppression of tumor growth [201]. The authors similarly demonstrated that incomplete RFA was also associated with CRLM progression and resistance to anti-PD-1 therapy in murine models [202]. While disappointing, the authors showed that combination therapy with complete RFA and anti-PD-1 therapy resulted in marked, durable reduction in tumor growth as well as increased infiltration of tumor-infiltrating lymphocytes [201]. 

**Table 1 cancers-13-06206-t001:** Summary of meta-analyses investigating 3-year PFS and OS after surgical resection or local ablation of CRLM and CRPM.

Modality	3-Year PFS	3-Year OS	Reference
CRPM			
Surgery	Not reported	68.6%	Zabaleta et al., 2018 [203]
RFA	Not reported	35–65%	Lyons et al., 2015 [204]
SBRT	13%	52%	Cao et al., 2019 [193]
CRLM			
Surgery	31.2%	63.8%	Beppu et al., 2012 [205]
RFA	24%	60%	Di Martino et al., 2020 [206]
SBRT	11.5 months *	31.5 months *	Petrelli et al., 2018 [207]

* Only median PFS and OS reported.

### 6.2. TME-Specific Targeted Therapy

#### 6.2.1. Amelioration of Inflammation

The CRC TME promotes a pro-inflammatory milieu, which has become a promising therapeutic target with repurposed agents. Aspirin, as an example, became the first drug recommended for chemoprevention based on compelling retrospective data [208]. Platelet activation is known to result in an immunosuppressive TME that spares cancer cells from immune surveillance. Aspirin inhibits platelet function and induces apoptosis of tumor cells by upregulation of TNFα and by inhibiting the mTOR pathway [209]. It also inhibits cell proliferation, metastasis, and angiogenesis through the inhibition of COX pathways [210]. Celecoxib, a COX-2 inhibitor, along with chemotherapy, demonstrated clinical benefit in COX-2-positive gastric cancer, but it did not meaningfully improve disease-free survival in combination with adjuvant FOLFOX for stage III CRC [211,212]. It is currently being evaluated in combination with anti-PD-1 immunotherapy in an ongoing phase II study of MSI-high/mismatch repair-deficient and high-tumor-mutational-burden CRC [213].

#### 6.2.2. Anti-Angiogenic Therapy

Anti-angiogenic agents impede the formation of new vasculature, which is essential for tumor growth and metastasis. The most well-studied anti-angiogenic agent is bevacizumab, a humanized IgG monoclonal antibody that targets and inhibits VEGF-A, which is prominently produced by tumor cells, TAMs, and other cells in the TME [69]. It is FDA-approved for mCRC [180]. In a meta-analysis of seven clinical trials comprising 1838 patients, Cao et al. demonstrated that the addition of bevacizumab to chemotherapy after primary tumor resection prolonged overall survival compared to chemotherapy alone [214]. Similarly, the phase III TRIBE2 trial found that bevacizumab + FOLFOXIRI resulted in superior median overall survival (27.4 vs. 22.5 months) and median progression-free survival after first-line treatment (12 vs. 9.8 months) [215]. The inhibition of angiogenesis in the TME may even increase the chance for resectability of CRLM; for example, the BECOME trial demonstrated that the R0 resection rate of initially unresectable *RAS*-mutant CRLMs was 22.3% in the bevacizumab + FOLFOX6 group, compared with 5.8% in the FOLFOX6-only group [216,217,218]. The efficacy of bevacizumab specifically for pulmonary metastasectomy is less understood. A case series of 11 patients who received neoadjuvant bevacizumab + FOLFOX/capecitabine followed by pulmonary metastasectomy had a 5-year disease-free survival of 10.9% [219]. A case report of rectal pulmonary metastasis treated with pulmonary metastasectomy as well as adjuvant S-1, irinotecan, and bevacizumab yielded a complete response [220]. It is possible that the presence of tumor cavitation predicts response to bevacizumab; in a study of 60 CRPM patients treated with bevacizumab + chemotherapy, the presence of tissue cavitation yielded a longer median overall survival (42.1 vs. 30.8 months) [221]. For those patients who progress despite bevacizumab + 5-FU-oxaliplatin-based chemotherapy, ramucirumab (anti-VEGF-R2 monoclonal antibody) + FOLFIRI has been shown to significantly increase OS and PFS in the phase III RAISE trial of 1072 patients and is now FDA-approved [181,222]. 

#### 6.2.3. Modulation of CAFs

Inhibition of CAFs is under investigation for mCRC in numerous clinical trials with mixed results thus far [154]. Targeting CAF activity through inhibition of the pro-desmoplastic Sonic Hedgehog pathway yielded no added clinical benefit in combination with FOLFOX or FOLFIRI + bevacizumab in a phase II trial of previously untreated mCRC [223]. Similarly, inhibition of LOXL2 via simtuzumab in combination with FOLFIRI yielded no clinical benefit in a phase II trial, and 90.6% of patients discontinued treatment due to progression [224]. CAFs are stimulated by and are a source of TGF-β. Thus, targeting TGF- β has garnered significant interest in CRC. A recent phase 1b trial of regorafenib with PF-03446962, a monoclonal antibody targeting TGF-β receptor activin-receptor-like kinase 1 (ALK-1) demonstrated significant toxicity and no clinical activity [225]. One of the emerging anti-TGF-β agents currently under clinical investigation is galunisertib, a small molecule selective inhibitor that suppresses the TGF-β pathway by inhibiting TGF-β receptor type 1 [226]. Galunisertib has been shown to reduce TGF-β-mediated suppression of T cells, provoking a strong cytotoxic T-cell response against CRC cells and enhancing the efficacy of ICIs [227]. In syngeneic CRC mouse models, galunisertib reduced tumor size, extent of peritoneal metastases, and CRLM, an effect that was dependent on CD8+ T-cell activity [162]. Clinical trials currently evaluating the efficacy of galunisertib or LY3200882 (another TGF-β receptor type 1 inhibitor) combined with capecitabine or 5-FU for advanced CRC, as well as neoadjuvant galunisertib with chemoradiation for rectal cancer, are underway [228,229,230]. 

#### 6.2.4. Repolarizing Macrophages

The conversion of macrophage phenotype from a pro-tumorigenic to an anti-tumorigenic phenotype is an attractive strategy. In a key study of therapeutic targeting of TAMs for CRLM, Halama et al. [231] demonstrated that cytokines, including CCL5 (ligand for CCR5) were highly expressed at the invasive margin. CCL5 was exclusively secreted by CD3+ T cells, which polarized TAMs (which contain CCR5) into a pro-tumorigenic phenotype. Treatment with maraviroc (a CCR5 inhibitor) in a CRLM tumor explant model induced macrophage-dependent tumor cell death [231]. These findings led the authors to launch a phase I trial in 14 mCRC patients treated with maraviroc, which resulted in a partial response in three patients, including a partial remission of CRPM. Histologically, all post-treatment CRLM samples showed a reduced proliferative index, tumor cell death, and a reduction in the expression of pro-tumorigenic, pro-angiogenic, and chemoresistance-promoting cytokines [231]. These promising results led to a phase I trial of maraviroc in combination with PD-1 inhibition for patients with microsatellite-stable mCRC, who experienced higher-than-expected overall survival but limited clinical activity, suggesting that further investigation is necessary [232]. 

#### 6.2.5. Inhibition of Tregs

While the impact of Tregs in CRC is controversial, the depletion of Tregs in CRPM may be clinically beneficial. In a murine CRPM model, Wang et al. [114] demonstrated that the administration of arsenic trioxide (As_2_O_3_) significantly reduced Treg infiltration and FoxP3 expression and reduced metastatic tumor size and number, and increased the cytotoxic activity of cytokine-induced killer cells [114]. A pilot study of 17 patients demonstrated that the addition of As_2_O_3_ to FOLFIRI for mCRC refractory to first-line FOLFOX/CAPOX yielded stable disease in 65% of patients, a partial response in 18%, and a median progression-free survival of 5.3 months [233].

#### 6.2.6. Augmentation of the Antitumor T-Cell Response

An emerging approach for CRC is chimeric antigen receptor (CAR) T-cell therapy, in which a patient’s own T cells are extracted, genetically modified to recognize tumor antigens, and reinfused. The application of CAR T cells in CRLM faces several key hurdles, including the immunosuppressive nature of the metastatic TME, the challenge of efficiently trafficking the cells to the tumor site, and the need to target the tumor-associated antigen [234,235]. In a preclinical murine model, Burga et al. [236] found that the regional infusion of anti-carcinoembryonic antigen (CEA) CAR T cells via the portal vein delayed the tumor progression of CEA+ CRLMs. However, the growth of CRLMs was associated with granulocyte colony macrophage stimulating factor (GM-CSF)-mediated expansion of the CD11b+ Gr-1+ myeloid-derived tumor suppressor cell (L-MDSC) population, which abrogated CAR T-cell proliferation within the tumor via STAT3-dependent PD-L1 expression [236]. Through the use of potent STAT3 inhibitors (e.g., JSI-124 or celastrol) combined with anti-GM-CSF therapy, the suppression of L-MDSC expansion improved CAR T-cell antitumor activity [236]. Furthermore, infusions of CAR T cells via the portal vein resulted in superior antitumor efficacy compared to systemic infusion [236]. These preclinical findings have been translated into ongoing and completed phase I and II trials investigating regional CAR T-cell therapy for CRLM [234,235]. 

Although they have established efficacy for other cancers, ICIs have only recently become FDA-approved for MSI-high/mismatch repair-deficient CRC [138]. This subtype is more receptive to PD-1 blockade [237]. The recent phase III KEYNOTE 177 trial of 307 patients with treatment-naïve metastatic MSI-high/mismatch repair-deficient CRC demonstrated that pembrolizumab was significantly superior to 5-FU-based chemotherapy (with or without VEGF (bevacizumab) or EGFR (cetuximab) blockade) with respect to progression-free survival (16.5 vs. 8.2 months) and was better tolerated [4]. Similarly, ipilimumab (CTLA-4 inhibitor) and nivolumab (another PD-1 inhibitor) have been approved for MSI-high/mismatch repair deficient mCRC [238,239]. Given their efficacy and favorable side effect profile, these agents are now being investigated in the neoadjuvant setting for early stage disease with a stage I trial demonstrating good tolerability and high rate of pathologic response (100%) in MMR-deficient tumors and moderate rate (27%) in MMR-proficient tumors [240]. Another combination of anti-CTLA-4/anti-PD-L1 blockade (tremelimumab/durvalumab) has also yielded improved overall survival in microsatellite stable cases in a recent phase II trial [241]. Notably, the administration of first-line FOLFOX (5-FU, oxaliplatin, and leucovorin) increased CD8+ T-cell infiltration and ameliorated CD8+ T-cell exhaustion in a CRC mouse model; combination FOLFOX with anti-PD-1 therapy significantly reduced tumor growth [242]. Thus, combination chemo-immunotherapy with FOLFOX and immune checkpoint blockade may be particularly promising for CRC and warrants further investigation in clinical trials. Additionally, as previously discussed, combinations with other ablative therapies, such as RFA, also warrant continued investigation. 

A summary of TME-specific targeted therapies currently under investigation in clinical trials for CRC is provided in Table 2. 

## 7. Conclusions

The TME of CRC is involved in virtually every step of tumor progression, invasion, immune evasion, and metastasis. The liver and lungs are the most frequently involved organs for CRC metastasis. While most research has focused on the primary tumor, the body of knowledge addressing the unique microenvironments in these organs and how their different cell types contribute to metastasis continues to evolve. There is a paucity of the literature specifically investigating the impact of TME components on CRPM. This may reflect the difficulty in obtaining adequate CRPM tissue for translational investigation, as image-guided biopsies may not yield adequate spare tissue, and surgical biopsies, even with minimally invasive thoracoscopic approaches, may place patients with advanced, unresectable disease at prohibitive risk. Further investigation in preclinical models of CRPM to better elucidate the unique features of the TME is, therefore, crucial. A clear observation in the CRC TME is that immune cells are important in cancer progression. The local TME may suppress anti-tumorigenic functions or modulate immune cells into a pro-tumorigenic phenotype. Abrogating pro-tumorigenic and augmenting anti-tumorigenic functions of these TME components has been demonstrated to enhance antitumor immune activity in preclinical models and clinical trials, but further investigation is warranted to maximize the benefit of novel immunotherapies. Indeed, while CAF-targeted trials in combination with chemotherapy have yielded mixed results, the efficacy of combination therapy with immunotherapy remains to be elucidated. A continued focus on how each of the individual components of the TME contributes to CRC progression and metastasis is critical for the determination of potential targets for novel therapies.

## Figures and Tables

**Figure 1 cancers-13-06206-f001:**
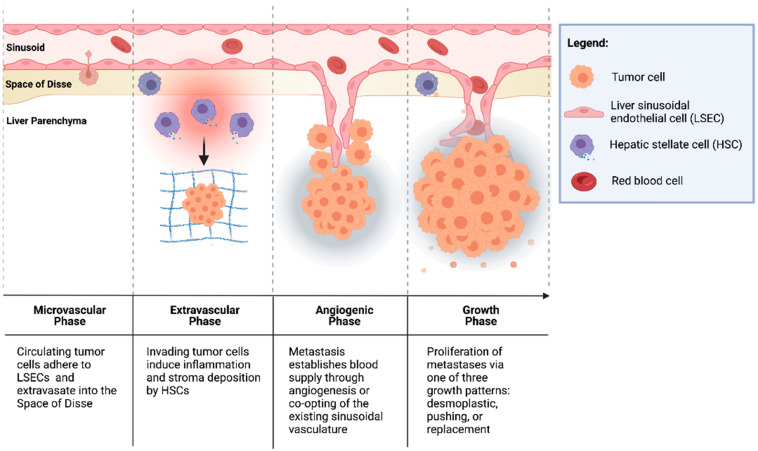
Formation of colorectal liver metastases through four overlapping phases: microvascular, extravascular, angiogenic, and growth.

**Table 2 cancers-13-06206-t002:** TME-specific targeted therapies currently under recent investigation in clinical trials. Red highlights reflect trials with negative findings. Green highlights reflect agents which are FDA-approved or recommended by the United States Preventive Services Task Force (Aspirin).

TME Target	Agent	Mechanism	Reference
Inflammation	Aspirin	COX inhibitor	Bibbins-Domingo et al., 2016 [208]
	Celecoxib	COX-2 inhibitor	Meyerhardt et al., 2021 [212]; NCT03638297 [213]
Angiogenesis	Bevacizumab	Anti-VEGF-A monoclonal antibody	FDA-Approved [180]
	Ramucirumab	Anti-VEGF-R2 monoclonal antibody	FDA-Approved [181]
CAFs	Galunisertib	TGFβR1 inhibitor	NCT03470350 [228] NCT02688712 [230]
	M7284	TGF-β inhibitor	NCT03436563 [196]
	LY3200882	TGFβR1 inhibitor	NCT04031872 [229]
	Simtuzumab	LOXL2 inhibitor	Hecht et al., 2017 [224]
	Vismodegib	SHH inhibitor	Berlin et al., 2013 [223]
	PF-03446962	ALK-1 inhibitor	Clarke et al., 2019 [225]
TAMS	Maraviroc	CCR5 inhibitor	Halama et al., 2016 [231]; Haag et al., 2020 [232]
Tregs	Arsenic Trioxide	Depletion of Tregs	Lakshmaiah et al., 2017 [233]
CD8+ T cells	Pembrolizumab	Anti-PD-1 monoclonal antibody	FDA-Approved [138]
	Nivolumab	Anti-PD-1 monoclonal antibody	FDA-Approved [239]
	Durvalumab	Anti-PD-L1 monoclonal antibody	Chen et al., 2020 [241]
	Ipilimumab	Anti-CTLA-4 monoclonal antibody	FDA-Approved [238]
	Tremelimumab	Anti-CTLA-4 monoclonal antibody	Chen et al., 2020 [241]

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
