# Peer review of "The Colorectal Cancer Tumor Microenvironment and Its Impact on Liver and Lung Metastasis"

_cancers, 2021, doi:10.3390/cancers13246206_

Round 1

Reviewer 1 Report

Metastasic process is the last step in cancer progression which carried out the dead of the patient in most cases. Also, the tumor miecroenvironment  is taken a relevant roll in the tumor progression and  it is being described its direct implication in the metastasic pregresión. Thus, the subject is relevant to the scientific community and the article is well written.  However it would be great if the author consider some suggestions:

  1. The contexts of figure 1 is very clear, however, the figure itself in case of  cell interpretation and location it is not totaly correct. The figure shows the liver sinusoidal endothelial cells (LSECs) in purple and in the image there are only 2 cells. What kine of cells are pink cells? if they are LSEC , they have to be purple, if not, it have to clarify. Also in case of hepatic stellate cells (HSC), it would be great if the figure have more than one HSC and some of them located in the space of Disse. 
  2.  In order to improve the compresion of the liver metastasic process, due to the  relevant implication of liver non parenquimal cells in the metastasis development it would be correct  if the author describe in more details its implication in the progression of the liver metastasis. 
  3. Althought the steps in colorectal liver metastasis is well described the lung metastasis is not very clear and it seems umbalanced. I suggest to complete it. 

Reviewer 2 Report

The review manuscript by Raghav Chandra and collaborators describes the role of tumor microenvironment in CRC lung and liver metastases as well as it describes the current treatment chances to target the tumor microenvironment and improve patients’ survival. The manuscript is well writing and presents many interesting data, however some concerns should be improved before publications in Cancers.

Main concerns:

  • I think that one of the main concerns is that the manuscript is too long. I think that there are a lot of information and the main objective of the manuscript is lost. For instance in the section 5 “The Tumor microenvironment in CRC” the authors described different roles of TME cells in primary and metastatic lesions. I think that just a general sentence should be enough for primary sites and a focused revision in metastases should be showed.
  • Sections 3.1.1/2/3/4: I would say that these metastases phase should be redacted for liver and lung metastases not just for liver metastases. I think that the organization of the manuscript should be changed regarding this issue.
  • Following the last point there is not a “The impact of the TME on colorectal liver metastasis” heading, in section 3 as it is for lung metastases in section 4. I suggest to homogenise the organization of the subtitles in section 3 and 4.
  • I think that extracellular matrix is also an important player in tumor microenvironment with an important role in CRC metastasis. I think that specific section regarding ECM should be added to section 5 and 6. There are many bibliography about its role and similarly, there are different clinical trials trying to modify ECM characteristics in order to abrogate CRC metastases.
  • In a similar way, there are not information about vascular cells (endothelial, lymphatic, pericytes… ) or mesenchymal cells in section 5.
  • Section 5 and figure 2 are difficult to contrast. First of all cites should be added to the figure and in this way it will be easy to compare the text and the figure information. Then in the figure there are information about primary and metastatic lesions. I think that some kind of separation between primary and metastatic information should be added (for instance, different colours, separation line…). Anyway, and as I previously stated, I think that this section is too large… Maybe the figure should just show the information about TME role in metastatic lesions.
  • Section 5 and 6 should be reorganized regarding cell type: immune cells (and then subtitles as macrophages, Treg…), stromal cells, vascular cells… I think it will help to the general organization of the manuscript.
  • Section 6.3.3. As far as I know there are much more information about CAF-TGFB targeting in CRC. I think that this information should be extended.
  • I think that the manuscript should include a more critical view of the futures perspective of the TME targeting highlighting the difficulties and challenge to target the TME.

Minor points:

  • The abstract and simple summary are almost identical… I think that the abstract might be larger and it should include more information about the different items of the review.
  • Line 38 and 191 are also almost identical. Please rephrase one of them.

Round 2

Reviewer 1 Report

The article summarized the implication of tumor microenvironment in the development of liver and lung metastasis from colorectal cancer.  The article is well writing and the subject is interesting for the scientific community. However, I suggest to the authors to revise the abbreviations. For example, in line 128 appear “liver sinusoidal endothelial cell” without abbreviation and in line 657 only LSEC.It should be corrected.

Author Response

Dear Sir/Madam,

Thank you so much again for your suggestions. We have updated several abbreviations (i.e. PD-L1, G-CSF, and LSEC) in the revised version of the manuscript. 

Thank you again for your time and consideration of our work.

Sincerely,

Raghav Chandra, MD

Reviewer 2 Report

Thank you for improve the manuscript following my previous comments. I think that the manuscript is now suitable for publication in Cancers.

Author Response

Dear Sir/Madam,

Thank you so much for your comments on our manuscript. We are very grateful for your suggestions to improve our submission.

Thank you for your time and consideration.

Sincerely,

Raghav Chandra, MD